# How does task structure shape representations in deep neural networks?

**Kushin Mukherjee**
Department of Psychology
University of Wisconsin-Madison
Madison, WI 53706
kmukherjee2@wisc.edu

**Timothy T. Rogers**
Department of Psychology
University of Wisconsin-Madison
Madison, WI 53706
ttrogers@wisc.edu

## Abstract

While deep convolutional neural networks can be trained to perform at human levels of object recognition and can learn visual features in the process, humans use vision for a host of tasks beyond object recognition, including drawing, acting, and making propositional statements. To investigate the role of task structure on the learned representations in deep networks, we trained separate models to perform two tasks that are simple for humans: imagery and sketching. Both models encoded a bitmap image with the same encoder but used either a deconvolutional decoder for the imagery task or an LSTM sequence decoder for the sketching task. While both models learned their tasks well, the sketcher model acquired representations that more clearly captured visual properties of the input, including location, size, and semantic category. This work suggests that acquisition of robust visual representations may depend importantly on the nature of the learning task.

## 1   Introduction

Cognitive science has long proposed that visual object representations are built from *units* or *features* that compose in complex ways to form a rich repertoire of possible percepts [Palmer, 1975]. Specific hypotheses about what the features are, and how they compose, have oriented around two views. Symbolic approaches [Biederman, 1987, Lake et al., 2015] suggest a finite, pre-determined vocabulary of visual primitives, and a visual "syntax" or rule-set for describing spatial relations amongst features. On this view, visual abstraction is possible because the object representations preserve essential characteristics of both the primitives and the spatial relations between them, but it remains unclear where the primitives and rules come from or how they might be learned. In contrast, deep convolutional neural network (DCNN) approaches view both the features and their composition as arising from learning about the statistical structure of the natural visual environment. For instance, DCNNs [Simonyan and Zisserman, 2015] trained on millions of real-world object images can assign photographs of objects to semantic categories with uncanny accuracy, and in so doing acquire a vocabulary of visual primitives that, in some ways, resemble response properties of neurons in the visual system [Kriegeskorte, 2015]. Such models suggest how visual features might be learned from visual input alone, but do not account for aspects of human visual perception that rely on componential representation–thus off-the-shelf DCNNs rely too much on visual texture, fail to correctly classify line drawings without additional training, and are susceptible to adversarial attack. Each of these failings suggests that standard architectures and training methods do not capture the part-based compositional processes that support human visual abstraction.

Perhaps this is unsurprising, since DCNNs are typically optimized to predict category labels for images. Human vision supports many other tasks, including drawing (i.e., visual communication;Fan et al. [2018]), speaking (i.e., propositional knowledge; Lambon Ralph et al. [2017]), and acting (i.e.,

praxic, haptic, and functional knowledge; Botvinick and Plaut [2004]). The structure of action in these tasks bears an important, componential relationship to the structure of objects–for instance, the sequence of strokes produced when drawing, say, a hand-saw relates in non-trivial ways to (1) the parts from which the saw is constructed, (2) the way we might grip a saw when using it and (3) the words we use when talking about the saw ("handle," "blade," "teeth"). Thus drawing, speaking, and using the saw all provide environmental cues about how the image should be partitioned into components, which in turn may aid in acquisition of the componential representations that support abstraction.

Recent approaches to learning visual representations have highlighted the need for sophisticated computational machinery, often combining neural network models with graphics-renderers. Generative Query Networks (GQNs) learn representations of 3D scenes without explicit labels through a probabilistic neural model, which can abstract away image-level details to represent the same scene from different viewpoints closely in represenational space [Eslami et al., 2018]. Adversarially-trained reinforcement learning models coupled with graphics engines have also been applied to teach agents to learn policies or 'visual programs', which can be thought to functionally approximate task structures as described earlier [Ganin et al., 2018] . Closer to the task we simulate in this paper, work by Gregor et al. [2015] applies recurrence to variational autoencoding models to capture the sequential nature of image production, although production in this work involved generation of bitmap images. While these approaches represent remarkable advances in ideas about image-generation, to our knowledge no prior work has considered whether or how different generation mechanisms might influence internal representations of the same visual object.

The current paper thus assesses how the nature of the output task influences the internal visual representations acquired by deep convolutional networks trained to reproduce an input image in model analogs of two different tasks: *imagery*, where the input bitmap is reconstructed over the outputs of a convolutional autoencoder, and *sketching*, where reconstruction involves generating a sequence of pen-strokes. Both models use the same convolutional encoder projecting to a flat densely-connected bottleneck layer. In the imagery network, these project to a deconvolutional decoder trained to reproduce the original bitmap. In the sketcher, the bottleneck layer projects to a two-layer LSTM trained to generate the associated sequence of strokes. Since the encoders are structured identically in both models, and both are trained on the same corpus of images, differences in the representations acquired by each must be due to the output task. We therefore used supervised and unsupervised techniques to assess the degree to which acquired representations in each model encode central elements of the input image including (1) its vertical and horizontal location on the image plane, (2) its size, measured as number of "on" pixels, (3) its category, and (4) its encoding of idiosyncratic details specific to a particular image.

## 2 Methods

### 2.1 Dataset

We created a simple 128x128 pixel *'Etch A Sketch'*-style drawing environment where every image was constructed through a sequence of horizontal and vertical lines. Thus each image can be represented either as a 128x128 bitmap tensor, or as a sequence of ( $\Delta x$, $\Delta y$, $p$ ) coordinates of length equal to the number of 'strokes' in a drawing. In this encoding, $\Delta x$ and $\Delta y$ captured the displacement of the pen in the horizontal and vertical directions (relative to its current position), while $p$ indicates a pen state (up or down) that determines whether a line should be drawn when the pen location moves [Ha and Eck, 2017]. We also defined functions that strung together successive strokes to describe simple shapes–right angles, arcs, quadrilaterals, and lines–that can combine in different spatial configurations to create drawings belonging to several semantic categories: tables, stools, chairs, mugs, briefcases, birds, sheep, dogs, lizards, and pigs. Members of each category had the same elementary parts arranged in the same relative spatial positions, but with parameters specifying the respective sizes of the corresponding part. For example, sheep could vary in their overall size, but also in the relative size of the head, length of neck, and leg length. The position of each drawing on the canvas was also sampled at random with uniform probability across horizontal and vertical dimensions. These routines provide a large universe of possible images, each the result of 5 independent factors: horizontal location, vertical location, size of the item, semantic category of the item, and the parameterizations that determine the relative size of each part. From the full universe we sampled 1000 drawings

independently with uniform probability for training samples, and another 1000 drawings as validation samples.

## 2.2 Model architectures

The autoencoder model took a 128x128x3 input bitmap tensor, passed through 3 layers of convolution and max pooling, and with all convolutional units employing rectified linear activations. The top convolutional layer was flattened and projected densely to a bottleneck layer of 512 linear units, which in turn was reshaped, deconvolved and upsampled to generate model outputs. We trained the model on 1000 samples using binary cross-entropy (BCE) loss for 150 epochs with batch sizes of 10, using PyTorch 1.6.

The sketcher model used the same convolutional encoding architecture through to the flat 512-unit encoding. This then projected to a 2-layer LSTM that produced a 3-element output across linear units for each of 20 timesteps (strokes) : a $\Delta x$ value, a $\Delta y$, and a $p$ value. This model was trained using mean-squared error (MSE) loss across LSTM output units for 40 epochs in batches of 10. To visualize the drawings produced by the sketcher, we fed the sequence of output values, together with a starting location, to our drawing program, which then generated a bitmap of the model sketch by moving a simulated 'pen' to the coordinates indicated by each output coordinate in turn, and printing a line with each move whenever the pen-state output was in the 'down' state.

## 3 Results

Figure 1 shows examples of model outputs for items not seen during training, for both autoencoder and sketcher architectures. Both models learned to produce outputs with low hold-out error and relatively good reconstructions, but with qualitatively different kinds of error: the autoencoder shows 'uncertainty' about which pixels should be 'on' in the area near the line segments, which produces the stippling pattern observed in the reconstruction. In turn, the sketcher exhibits uncertainty about the exact location of line endings, which produces the slight misalignment of strokes in the image reconstruction. Which provides a better reconstruction of the image? Since both the nature of the outputs and the loss function differ between models, they cannot be compared solely on reconstruction loss. We instead measured the *perceptual similarity* between the input image and each reconstruction, following the shape-matching method described in Belongie et al. [2002]. Their shape-matching cost is invariant to rotation, shear-transforms, and size differences between images so we take it as a good measure of perceptual similarity between any given pair of objects. Because this is an algorithm with a high time-complexity, we took a small subset of 45 validation images like the ones in 1 and computed the average shape-matching perceptual similarity between the ground truth input and the reconstructions from the autoencoder and sketcher models. In our setup, maximum similarity corresponds to a shape-matching cost of 0 and the minimum similarity can be arbitrarily large because the cost is a summation of $n$ $\chi^2$ test statistic values, where $n$ is the number of points sampled from the image. The mean matching cost for autoencoder reconstructions was $10.55$, $(sd = 5.18)$ and sketcher reconstructions was $13.27$, $(sd = 4.29)$. This shows that both models reached comparably high levels of perceptual fidelity when reconstructing the input bitmaps.

The central question is whether these differences in output task produce systematic differences in how each model encodes a given input image. In particular, does the output task lead to systematically better or worse representations of visual information in the image? To answer this question, we analyzed the internal representations generated over hidden units for 400 images unseen during training from 4 of the 10 image categories (chairs, stools, dogs, sheep): 100 sampled randomly with uniform probability from each category, and situated randomly with uniform probability across the image canvas. For each architecture we considered (1) what image properties dominate the similarity structure of the model's representations, as observed in a low-dimensional embedding? (2) What image properties can be reliably inferred from each model's learned representations using linear decoders? (3) What blend of image properties most strongly influence a model's representations, as assessed using representational similarity?

*Embeddings of learned representations.* To understand which image factors strongly influence learned representations, we used classical multidimensional scaling (MDS) to compute 2D embeddings of the representations for 400 held-out images in each model. Figure 2 shows the results: representations in the sketcher are clearly organized to reflect the horizontal and vertical locations of the sketch on the

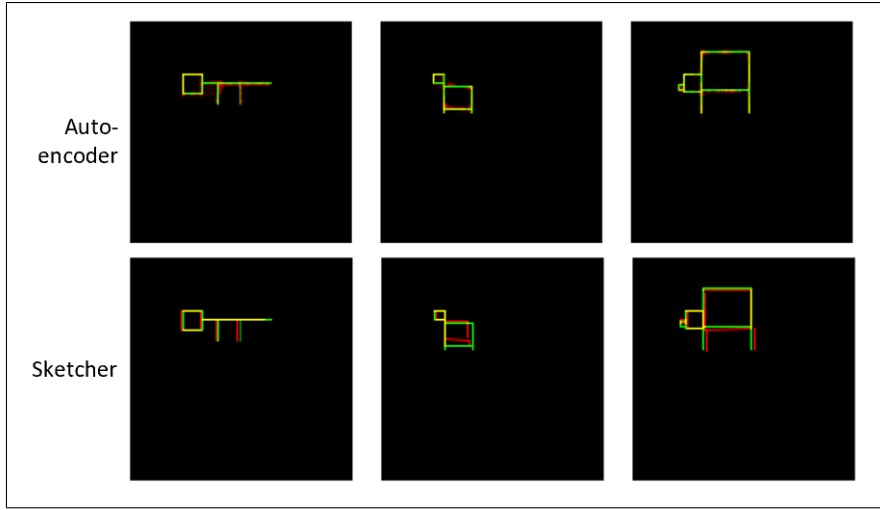

Figure 1: Examples of model outputs reconstructing a lizard (left), sheep (middle), and pig (right). The input image is shown in the green channel while model reconstruction is shown in the red channel, so yellow indicates pixels that are 'on' in both input and reconstruction. Both models can generate approximately correct reconstructions for unseen images, but the nature of the errors differs: autoencoders show uncertainty about which pixels should be on near the image components, while sketchers may be slightly off in the start and end points of the image components.

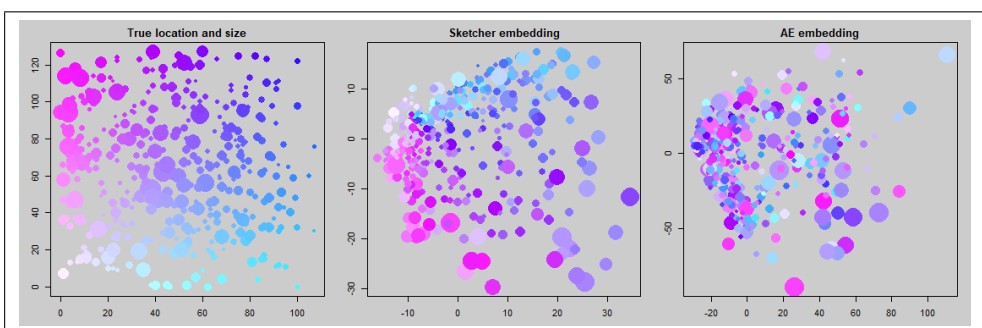

Figure 2: 2D embeddings of model representations. The left panel shows the true horizontal and vertical locations of each test image and how these are colored in the remaining plots. Circle size is proportional to the number of 'on' pixels in the image. Middle and right panels show the corresponding 2D embedding of model representations for the same images, in the sketcher and autoencoder models respectively. The horizontal and vertical image locations clearly organize the embeddings for sketcher representations but not the autoencoder.

image canvas, while also weakly reflecting image size. None of these features are clearly apparent in embeddings of the autoencoder representations, where no pattern is easily discernable. Similar results were observed using nonlinear embedding techniques such as t-SNE.

*Decoding image information with linear classifiers.* Autoencoder representations may still effectively encode important image properties despite the seeming disorganization of 2D embeddings for this model. To assess this possibility, we applied linear regression/pattern classification to decode image properties from the learned representations for each model. We used linear regression with elastic-net regularization to predict, from an image's learned representation, each of its continuous properties (horizontal and vertical location). For category decoding, we fit a four-way multinomial classifier regularized with the elastic net to predict the category label of the drawing from the learned representations. Note that the only category information available to the model during learning was that objects from the same category have a similar sequence of generative actions, and hence are composed of similar parts arranged in similar spatial configurations–no category labels were provided in training. For each model we applied 10-fold cross-validation and computed the mean error across

Table 1: Decoding of image properties from hidden representations

| Model | Image property | | | |
| | Category (accuracy) | Horizontal location ($r$) | Vertical location ($r$) | Size ($r$) |
|---|---|---|---|---|
| Autoencoder | 36.50% | 0.71 | 0.61 | 0.64 |
| Sketcher | 54.00% | 0.91 | 0.81 | 0.87 |

Table 2: Correlations between model features distance matrices and image property distance matrices

| Model features | Image property | | |
| | Location ($x, y$ coordinates) | Size | Shape-match |
|---|---|---|---|
| Autoencoder | 0.39 | 0.001 | −0.009 |
| Sketcher | 0.49 | −0.026 | −0.011 |

hold-out items from each fold. The results are shown in Table 1, and are unambiguous. While was possible to decode all the image properties better than chance from AE representations, decoding accuracy was much higher for representations learned by the sketcher.

*Representational similarity.* The preceding results show that both models learn representations that express some important image properties, but the sketcher more systematically encodes all properties assessed (location, size, and category). We next applied representational similarity analysis (RSA) [Kriegeskorte et al., 2008] to assess whether the model representations differ in their sensitivity to different intrinsic image properties, including location, size, and shape [Belongie et al., 2002]. Specifically, we computed a *representational-dissimilarity matrix* (RDM) using the 512-dimensional encoding of the 400 test images from each model. We also computed *target dissimilarity matrices* (TDM) for each type of image property. For location, this was the Euclidean distance between image centers in the $(x, y)$ image plane; for size, it was the Euclidean distance between the flattened bitmaps such that on pixels were 1s and off pixels were 0s; and for shape it was the alignment metric of [Belongie et al., 2002] described previously. Finally, we computed Pearson's $r$ between the upper triangle each model's RDM and each TDM. The results appear in Table 2. Representations in both networks are most sensitive to the location of the object within the canvas relative to the size and shape of the object. It is worth noting that the sketcher model produces outputs in an egocentric coordinate frame to which we later add a starting location to generate an output image. Nevertheless, the sketcher is *more* sensitive to location information than is the autoencoder, whose output relies on turning on the correct pixels in the correct locations.

*Training models with fixed image location.* In both models, representations are largely dominated by the location of the image on the input plane. In some ways this is unsurprising, since both models must learn to generate correct outputs from training data in which the same item generates largely or completely non-overlapping inputs when it appears in different spatial locations. Do the representational differences we have described arise solely because the two models differ in how they learn to solve this problem? To answer this question, we again trained both models to reconstruct 1000 images, but with all inputs centered on the same location in the input plane. Images still varied in overall size, the sizes of their respective parts, and the spatial configuration of their parts, but were centered on the same $(x, y)$ location. The training process was exactly the same as described in section *2.2*, except with this new sample of training images. We then computed internal representations generated for 400 novel inputs, equally sampled from the categories *dog*, *sheep*, *chair*, and *stool* as in the previous analyses. The central questions were whether and how the architectures differed in their ability to represent size and category structure.

We first applies an unsupervised approach by computing and inspecting 3D embeddings of the representations using classical MDS. The results are shown in Figure 3. The image category is apparent in both models, but the categories appear to be more clearly separated in the sketcher embedding. Image size (measured as number of "on" pixels) appears to be captured moderately for the animals, encoded almost linearly along z-axis of the sketcher embeddings and along a nonlinear manifold in the auto-encoder embeddings. Size information for the chairs and stools is not clearly expressed in either embedding.

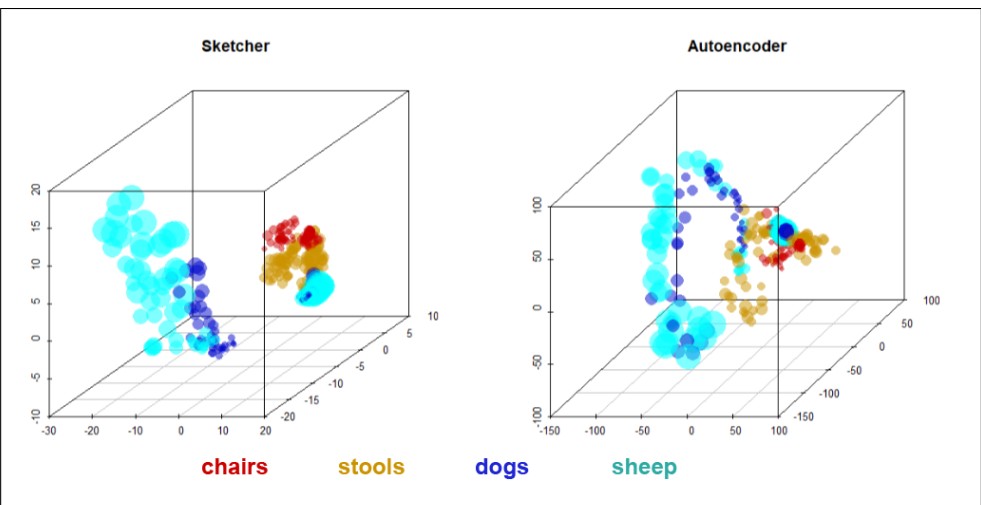

Figure 3: 3d embeddings of internal representations for models trained on images always appearing in the same location. Dot color indicates image category, while dot size indicates the number of "on" pixels in the image. Image category is evident in both embeddings, but categories seem better-separated in the sketcher. Both models appear to capture image size for animals but this is less clear for chairs and stools.

To more quantitatively compare the degree to which each set of representations express category and size information, we again applied supervised techniques. For category information, we fit a 4-class multinomial classifier, regularized with elastic net, and evaluated for performance using 10-fold cross-validation. Classification accuracy on held-out sets was near-perfect for the sketcher, which achieved 99.8% accuracy. Accuracy for the auto-encoder was still quite good, but reliably worse, at 92% on average across hold-outs. For size, we fit linear regression models to predict the number of "on" pixels from model representations, again applying the elastic net regularization and evaluating performance on held-out items using 10-fold cross validation. The sketcher showed a mean $r^2 = 0.9$ predicting the image size for held-out items, indicating that size is remarkably well encoded by a linear combination of learned features. The auto-encoder did significantly less well, showing a mean $r^2 = 0.74$ predicting sizes of held-out images.

Together these results suggest that the differences between architectures shown in preceding analyses did not arise solely from the ways the two models learn to handle variation of image location in the input plane–even when location is held constant, the sketcher acquires representations that better capture important image properties, including semantic class and size.

## 4  Discussion

This paper provides a proof-of-concept of the importance of output modality in shaping the structure of object representations in a set of simple yet naturalistic visual tasks — imagery and sketching. We show that while both models are able to learn their respective tasks quite well and produce reconstructions that are perceptually similar to their target, the sketcher model learns representations (1) in fewer training cycles that are also (2) better able to decode properties of the targets. These properties include a latent category structure that is defined in terms of the sequence of actions needed to produce the target. Information about category structure is also available to the autoencoder insofar as objects belonging to the same category also 'look' similar save for placement on the canvas and the size. Nevertheless, even with many more training cycles, the autoencoders ability to decode category structure remains inferior to the sketcher. Thus, the specific output modality of generating sequences of pen strokes is a critical factor.

This highlights the benefit of having multiple kinds of cues in learning visual object representations. This work is supported by findings from cognitive neuroscience and neuropsychology [Lambon Ralph et al., 2017, Lambon Ralph, 2014] and adds to a growing literature on how modern machine learning models can be augmented with theory from the cognitive sciences [Lake et al., 2017].

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
