# OpenReview forum: "How does task structure shape representations in deep neural networks?"
_NeurIPS.cc/2020/Workshop/SVRHM — SVRHM@NeurIPS Poster_

### Official Review · AnonReviewer2 · 2020-10-27
**Interesting preliminary work...**

**Rating:** 7
**Confidence:** 4

**Review:**

Summary
-------

This paper explores how different tasks affect the latent representation learned by a network. The authors train two networks: a fairly standard convolutional autoencoder, and, an encoder-decoder model where the encoder is the same as the AE, but the decoder is a two layer lstm which produces coordinates to move a pen to from the current position, as well as the pen state (up/down). A dataset of drawings is generated together with the corresponding pen-stroke paths and used to train the models. The authors then explore what information is captured in the representational vectors produced by the encoder parts of the two models. They find that although both representations contain information about the underlying variables that generated the input, the sketcher model's representation has a significantly stronger (and perhaps more disentangled) representation than the AE.

Positives
---------

- Obviously very preliminary work, but a good paper overall with interesting initial findings.
- Well written and easy to follow


Questions/Concerns/Comments
--------

- Can the dataset (and code to generate it) be made public?
- Have the authors looked at similar work investigating how differing tasks affect what a model learns (for example I saw this at one of the NeurIPS workshops last year: https://arxiv.org/pdf/1911.05546.pdf)
- It's probably also worth looking recent work on disentanglement within models - it strikes me that the sketcher learns a more disentangled representation of the underlying features that control the image generation than the AE does.
- Finally, I'd encourage the authors to revisit the encoder network structure (in future work!) - the max-poolings probably make it quite hard to learn about specific spatial information with high resolution (which could explain the sensitivity); perhaps augmenting the network with spatial information (https://arxiv.org/abs/1807.03247) could help?


Rationale for score
-------------------

Overall, I think this would make an excellent poster at the workshop and look forward to talking to the authors about the work.

---

### Official Review · AnonReviewer1 · 2020-11-01
**accept**

**Rating:** 8
**Confidence:** 5

**Review:**

The authors investigate the role of two types of task structure (imagery and sketching) on the learned representations in deep neural networks. The paper is well-written, the proposed modeling of the two tasks (convolutional autoencoder, encoder-LSTM decoder) is interesting and novel. I recommend accepting the paper.

---

### Official Review · AnonReviewer3 · 2020-11-01
**Reviewer 3**

**Rating:** 6
**Confidence:** 3

**Review:**

The paper analyzes how difference in the way the task is specified can affect the representation learned by the latent variables of an auto-encoder. In particular, an auto-encoder is trained to reconstruct simple stylized images using either a convolutional decoder that outputs raw pixel values, or an LSTM decoder that outputs a sequence of pen strokes. The results show that both models can create a sufficient representation of the input image (the reconstructions are qualitatively good), however while the stroke-based encodes linearly position information.

The question proposed by the paper — how the ability of an agent to act and communicate in an environment shape its learned representation — is indeed interesting and impactful. The paper tackles the setting where an agent is tasked with reproducing an image using two different modalities, for which it introduces a simple toy dataset to test the difference.

It would be interesting to extend these results to more complex tasks, and also to discuss the relation similar setup described in the literature. For example, http://proceedings.mlr.press/v37/gregor15.pdf describes auto-encoding of an image in multiple steps using an attention mechanism that resembles drawing, in which case they observe a more semantic representation emerging. Also https://arxiv.org/abs/1804.01118, https://arxiv.org/pdf/1910.01007.pdf describe image generation using a stroke based generative model, although they do not focus on the qualities of the learned representation. https://science.sciencemag.org/content/360/6394/1204 notes that when a network is tasked to imagine the scene pictured from a different point of view it learns a representation of the input image that is more semantic.

---

### Decision · Program_Chairs · 2020-11-02

Accept (Poster)